



# Technical note: Stage and water width measurement of a mountain stream using a simple time-lapse camera

Pauline Leduc [1], Peter Ashmore [1], and Darren Sjogren [2]

[1]University of Western Ontario, Department of Geography, London, Ontario, Canada
[2]University of Calgary, Department of Geography, Calgary, Alberta, Canada

*Correspondence to:* Pauline Leduc (pleduc3@uwo.ca)

**Abstract.** Remote sensing applied to river monitoring adds complementary information useful to understand the system behavior. In this paper we present a visual stage gauging and width measurement method using a ground-based time-lapse camera and a fully automatic image analysis algorithm for flow monitoring at a river cross-section of a steep bouldery channel. The remote stage measurement was coupled with a water level logger (pressure transducer) on site and shows that the image-based
method gives a reliable estimate of the water height variation and daily flow record when validated against the pressure transducer (R = 0.91). From the remotely sensed pictures, we also extract the water width and show that it is possible to extract correlation between water surface width and stage. The images also provide valuable ancillary information for interpreting and understanding flow hydraulics and site weather conditions. This image-based gauging method is a reliable, informative and inexpensive alternative or adjunct to conventional stage measurement especially for remote sites.

## 1   Introduction

Conventionally river discharge is gauged using continuous measurement of stage (typically, at temporary sites, using a pressure transducer and data logger) that is converted to continuous discharge data using a stage-discharge curve established for the site. In some cases installation of a stage recorder is problematic and in complex flows interpretation of stage fluctuations may be uncertain. These conditions may arise, for example in steep, bouldery or rock bed channels. Image-based measurements may

provide equivalent data to the pressure transducer record while giving additional information such as water width, state of flow, water surface configuration and indications of flow hydraulics. For larger rivers, satellite or aerial images may provide useful stream gauging data (e.g. Smith et al. (1996); Gleason and Smith (2014)) but for small streams and very high frequency (minutes) over extended periods, satellite and airborne platforms do not provide sufficient resolution or temporal frequency (Gleason et al., 2015). Ground-based remote sensing increasingly is providing a wide range of data for many applications for

monitoring river flow and morphology especially in smaller channels or where high frequency data are needed for extended time periods (Bertoldi et al., 2012; Wheaton et al., 2013; Javernick et al., 2014; Gleason et al., 2015).

Remote sensing based on photogrammetry technology provides an efficient topographic tool and access to topography and hydraulic characteristics (Javernick et al., 2014). However, the large amount of data needed to generate topography make it difficult to apply on small streams with a high sampling frequency (Gleason et al., 2015).





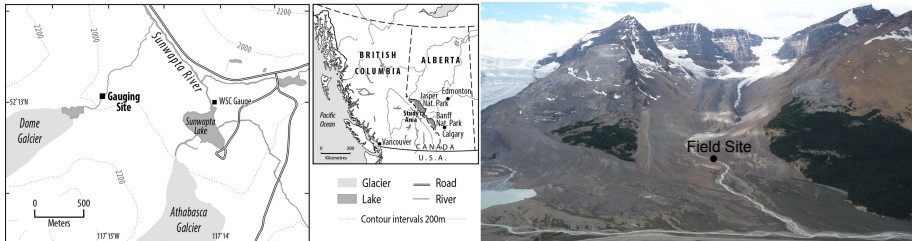

**Figure 1.** Map of the site location (left) and view of the Dome stream (right), image attribution - J.T. Gardner

In relation to flow characteristics, water surface width can be measured from ground-based cameras (Ashmore and Sauks, 2006; Gleason et al., 2015) and correlated with discharge to establish a width-discharge curve in some types of rivers, and local flow velocity has been measured using particle image velocimetry (Creutin et al., 2003; Hauet et al., 2008; Tsubaki et al., 2011; MacVicar et al., 2012). Direct measurement of stage is less well developed although Young et al. (2015) obtained a water
level and discharge record using manual image processing on a small, steep channel using inexpensive ground-based cameras combined with known channel geometry and roughness assumption. A more automated method that does not require manual image classification and channel geometry and hydraulic assumptions would be useful.

Methods for automated image selection and measurement are also needed in order to process $10^3$ or $10^4$ images that may come from high frequency time-lapse RGB imagery (Gleason et al., 2015). Here we test a simple time-lapse cameras system
for directly measuring stage and water surface width using image classification, and develop automated image selection and classification processes that retain a much larger proportion of the images than the process described by Gleason et al. (2015). We apply the method to monitor flow in a steep, boulder glacier-fed mountain stream which presents challenges for any form of flow gauging.

## 2   Measurement method

### 2.1   Site

The study site is located on a small, steep, bouldery reach of a stream, approximately 100 m downstream of the outlet from the small pro-glacial lake of the Dome Glacier, in Jasper National Parker, Alberta, Canada (Fig. 1). Site elevation is about 1800 m above sea level and the upstream drainage area is primarily the subglacial drainage of the Dome Glacier which is about 3 km in area. The stream is a left bank tributary of the Sunwapta River and the primary larger purpose of the study is to better estimate
the total discharge of a braided section of the river downstream of the Sunwapta–Dome Glacier tributary confluence by directly monitoring the Dome Glacier streamflow during the summer meltwater flow season. The Water Survey of Canada gauging station on Sunwapta River, at the outlet of the pro-glacial Sunwapta Lake, a few hundred meters upstream of the confluence with the Dome Glacier stream (Fig. 1) provides detailed discharge records for the braided reach but does not account for the tributary contribution. Ashmore and Sauks (2006) measured water surface width from oblique time-lapse oblique images on the





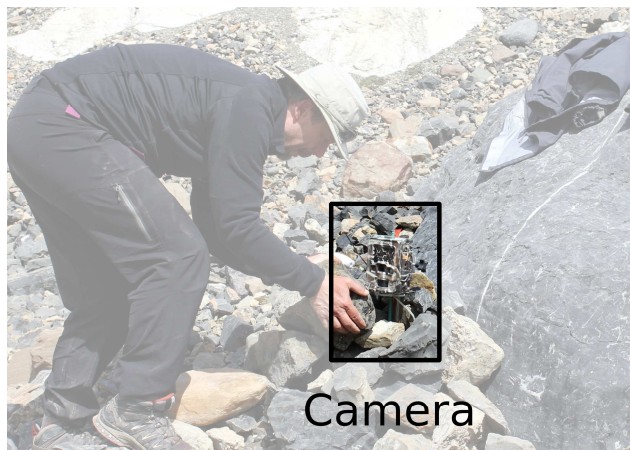

**Figure 2.** Camera set up. The camera is set on the right bank of the stream, a few meters above the water level. The camera is clamped to a pole hammered into the ground.

braided reach of Sunwapta River downstream of this tributary and established a relationship with discharge at the Water Survey of Canada gauging station using a small number of gauging measurements in the braided reach. But continuous measurement of the discharge of the Dome Glacier stream has not previously been used for monitoring this narrow, steep tributary to directly measure its contribution, daily flow variation and timing of daily peak flow relative to the Sunwapta River discharge. The
stream flow is mainly controlled by snow and glacier melt in summer producing a regular diurnal hydrograph with long-wave changes due to average air temperature and synoptic weather conditions in the summer. A straight, single thread reach of the channel was chosen for the gauging location.

## 2.2   Field setting

The main objective of the study was to use ground based remote sensing to measure the flow characteristics (flow stage and
water surface width) and peak flow periods in the daily flow cycle in this pro-glacial stream and assess the flow and timing of peaks relative to the flow of Sunwapta River. Standard pressure transducer measurement of stage is possible at this site (and is used here for comparison with image-based measurements) but we are interested in testing whether reliable image-based measurements are possible to complement or replace stage-only data with water level, water surface width and state of flow information from remote camera monitoring. A Reconyx Hyperfire camera was set on the right side of the reach (Fig. 2),
clamped to a pole hammered into the rocky ground, facing an almost vertical face of a large boulder on the opposite bank of the river. The entire stream width is visible in the pictures (Fig. 3).

Pictures were taken every 15 minutes during daylight (typically 6 a.m. – 10 p.m. at this location in the summer and the daily peak is usually 4 p.m. –7 p.m. ) which corresponds to the sampling interval and timing of the gauging station on the Sunwapta River. During the study period, from June 13 [th] to September 22[nd] 2015, 7284 pictures were taken, saved on the SD card and
downloaded at the end of the study period. Two stage boards were installed one on each side of the stream (Fig. 3). On the left





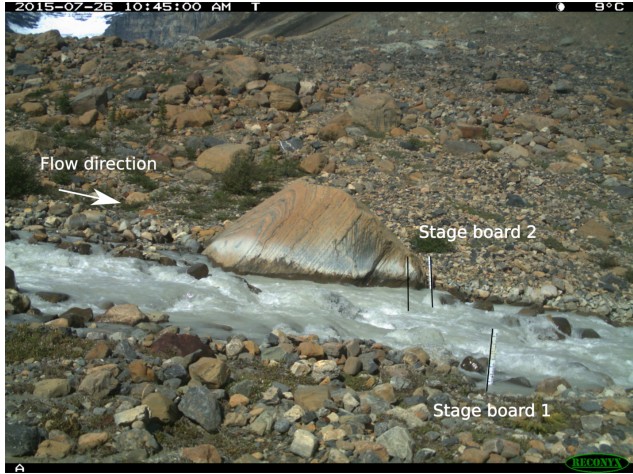

**Figure 3.** Reconyx raw picture showing the installation of stage boards, pressure transducer and boulder used for gauging (center of picture). Black lines represent the 3 different tested profiles.

bank a water level logger was installed in a vertical pipe in the stream bed next to the stage board with stage recorded at 15 minute intervals throughout the study period for comparison with the image-based stage data. Level data were compensated for atmospheric pressure. The boulder has an almost vertical surface, facing the camera and it was also calibrated for stage measurement.

In the rest of the paper, the phrase "transducer data-set" and the notation $H_{transducer}$ correspond to the stage coming from the pressure transducer and the phrase "camera data-set" and notation $H_{camera}$ correspond to the stage coming from the image analysis.

## 3   Stage and water width measurement

### 3.1   Picture quality

A goal of this method is to minimize any manual treatment of the images to select an analysis set of images and to estimate water stage and water width from those images. Consequently a screening treatment was applied to remove unusable pictures prior to analysis. The initial RGB picture size is 1536*2048 pixels, which was saved in .jgp format and converted into grey scale. Dark pictures corresponding to twilight were identified using a very low standard deviation of the grey intensity of the picture and automatically deleted from the data set. Over the summer, weather conditions also negatively impacted image

quality. During rain or snow episodes, the images are blurrier and water drops on the camera block the view. Snow cover on the ground is a lighter shade than the water, the opposite of the normal weather conditions. In the late afternoon and evening, the sun shines into the camera and induces two kinds of issues. First, sun light directly hits the camera and the picture is almost



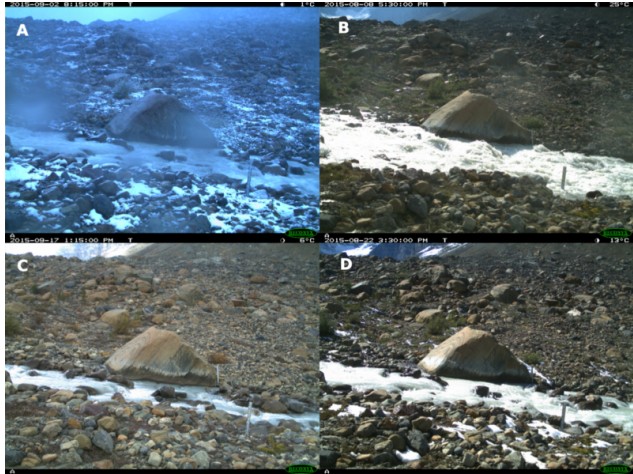

**Figure 4.** Images illustrating the main issues affecting image quality and measurements during the stage and width calculation: (a) heavy weather conditions (b) boulder shadow and intense water reflection (c) emerged rocks on the main channel at low flows (d) light snow cover on the edge of the stream.

entirely saturated. Second the water surface is saturated by reflections and the boulder facing the camera creates a large dark shadow on the water surface. Each of these issues interferes with the image processing and had to be compensated for.

For all the previous reasons (rain, drops on the lens, snow, sun effects) and as mentioned in Gleason et al. (2015) and Young et al. (2015), pictures have to be sorted. We developed automatic sorting processes and set two different output options : 1. the

image is removed from the data set or 2. The image is retained but different processes of classification and detection are used for particular conditions (see section 3.2).

## 3.2    Picture sorting

Image quality issues arise throughout the process of water detection and width estimation : poor weather conditions (Fig. 4 (a), shadows (Fig. 4 (b), emerged rocks in the stream (Fig. 4 (c) and light snow cover on the edge on the stream (Fig. 4 (d)). To deal

with those four issues, four different selection tests based on different target zones in the images were applied before or during the stage and water width measurements. All four tests were applied to the grey scale pictures and in the following descriptions standard deviation or averages refer to calculations on pixel intensity.

The first test was made to remove pictures taken under bad weather conditions (heavy snow or rain episodes, and sunlight directly into the camera) and it is based on the rocky zone target on the left of the picture (area 1 on Fig. 5). The standard devi-

ation of this surface is high due to the apparent roughness coming from the rocky surface. Under adverse weather conditions, that area is smoother and the standard deviation drops (snow cover with normal weather condition is not included in that test because even with snow, the roughness from the block elevation makes the standard deviation high enough) and the picture was removed from the data sets. Over the 6717 pictures, 12% were removed after this test (Fig. 6).





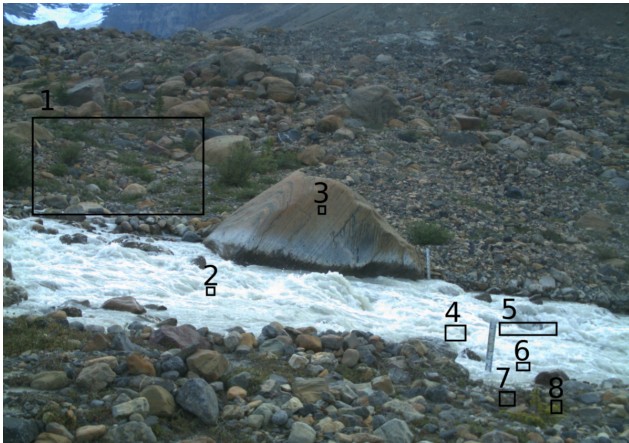

**Figure 5.** The eight different targets zones used in the picture sorting process.

During the water surface detection, a significant boulder shadow sometimes interfered with the calculation. Figure 5 shows the target zones 2 and 3 that were used to detect the boulder shadow, combining a height reflectance of the water (on zone 2) and a dark zone on the boulder face (on zone 3). A different water detection threshold was applied for those pictures (see section 3.3).

Two main issues interfering with the water edge detection arise: the rocky bottom of the stream (Test 3) and the snow on the river banks (Test 4). On Figure 5, target zone 4 is used as a water reference, this part of the stream always had water even at a very low stage. Target zone 5 and 6 are located where rocks are emerged at low flows. The mean value of both target zones is compared to the mean value of the reference zone, the target value less than half the reference zone corresponds to submerged rocks. The threshold based on half the value of the reference zone was set empirically after going through a substantial portion

the data set. As described in section 3.6, width estimate is based on cross sections and the profile with emerged rocks is re-sized based on the rock emergence/submergence.

The snow cover on the edge of the stream is detected using Test 4 and target zone 7 and 8. The test is based on the lighter color of the snow, mean values on zone 7 and 8 are compared to the mean value of the picture, and snow cover corresponds to brighter values on the target zone. As for the case of emerged rocks, the cross section used for the width detection is re sized.

The 4 tests are summarized in Table 1.

### 3.3 Water level

Stage was measured by detecting the water surface line on the images. Instead of a global, manual approach using edge detection (Young et al., 2015), we based the analysis on local site conditions. Black lines on Figure 3 on both stage boards and the vertical surface of the boulder represent each location where grey-scale profiles from the images were extracted to detect

the transition from water to stage board or boulder surface in the image (Fig. 7) and so locate the water surface in image space.





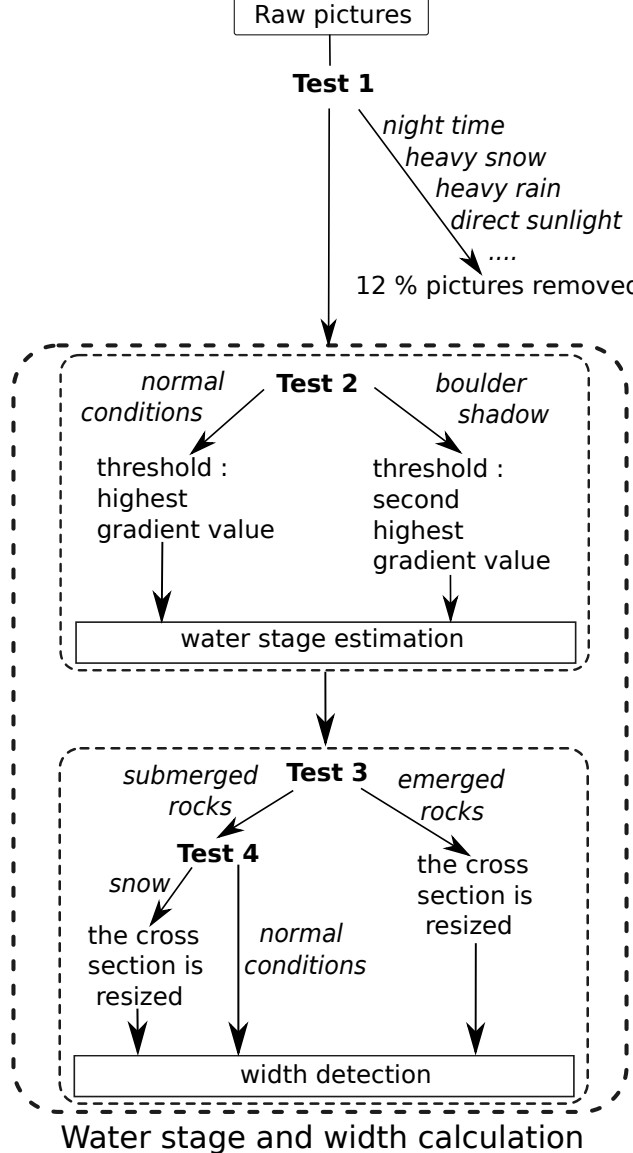

**Figure 6.** The picture sorting process: four different tests are used to remove heavy weather conditions or night pictures from the data-set (Test 1), to detect pictures with an important boulder shadow (Test 2), or emerged rocks on the main channel at low flow (Test 3) and finally a light snow cover on the edge of the stream (Test 4).

The water and boulder/stage board transition signal is clear for each water level location. On both stage boards the image signal is smoother on the water than on the board itself (Fig. 7). However, the transition between rock and water is more obvious on the boulder site (Fig. 7 (c)) than on the left bank stage board (Fig. 7 (b)). On the board located on the right bank of the stream the transition between the board and the water is not as clear and the flow stage rise is more difficult to detect. Furthermore, at



| Test number | Target area (Fig. 4) | Picture issue | Conditions (the test is failed if the condition is fulfilled) | Action if test failed |
|---|---|---|---|---|
| 1 | 1 | Night time, heavy rain, snow... | Standard deviation lower than 20 | Removed from the data set |
| 2 | 2 - 3 | Boulder shadow | Area 2: mean value higher than 235 - indicating almost direct reflection on the water surface Area 3: mean value lower than 95 - indicating a dark area on the boulder surface | The water detection is based on the second highest gradient point instead of the first |
| 3 | 4 - 5 - 6 | Emerged rocks on the main stream | Area 4 is the water colour reference value Area 5 and 6: the mean value is less than half of the reference value | The width calculation profile is re-sized |
| 4 | 7 - 8 | Light snow cover on the edge of the stream | Area 7 and 8: the mean value is higher than the average of the entire picture and the standard deviation of both area is less than 50 (indicating a smooth area) | The width calculation profile is re-sized |

**Table 1.** The 4 different picture sorting tests. Threshold are set using both particular and normal conditions pictures

low stage the bottom of the scale emerged above the water and the measurement was then impossible. Consequently this stage board was removed from the analysis. Without the boulder we would have used the remaining stage board as the water stage estimation but in this particular site the stage board on the far side is used as an independent visual check because the boulder surface gave a clearer signal.

5    On the boulder the water transition corresponds to an obvious inflection point in the image intensity (Fig. 8 (a)) and a local peak in the gradient of the smoothness profile (Fig. 8 (b)). The inflection point is detected using two conditions. The first, on the gradient profile, was used to pick high gradient values. The second condition was based on a grey shade threshold so that only the lowest values in the grey-scale profile are considered since higher values represent the rock becoming darker when it is wet. Using this combined method, the water line position in pixel coordinates can be automatically detected for each picture.

10  In pictures with the boulder shadow issue, the second highest gradient point is considered instead of the first.



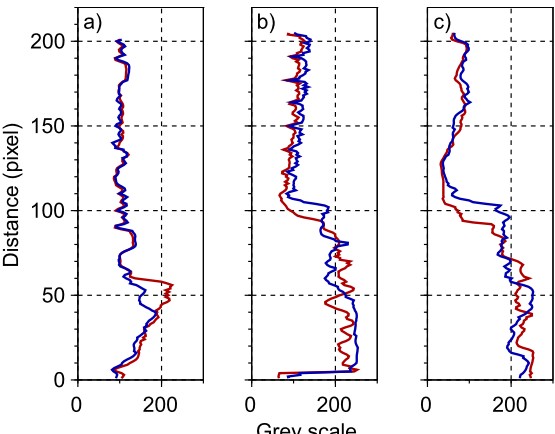

**Figure 7.** Grey-scale profiles for water surface measurement. (a) the first stage board on the right side of the stream, (b) the second stage board, on the left side of the river, and (c) on boulder vertical surface.

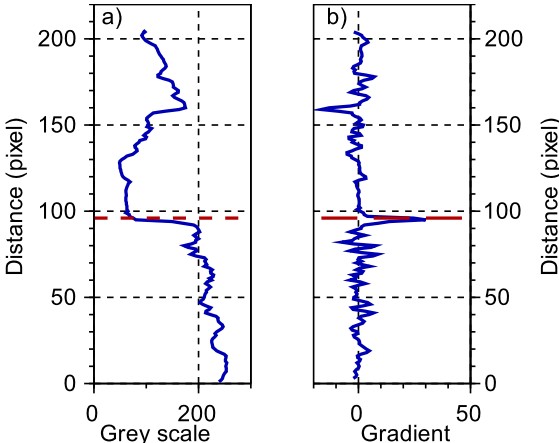

**Figure 8.** The boulder gauging station profile. The dashed lines represent the water surface. Water is located below those lines. (a) the grey-scale profile, and (b) the gradient of the profile, the calculation step is 2 pixels. The gradient plot makes the inflection point and the water level detection easier.

## 3.4 Water depth calibration

Given that the boulder surface is almost vertical, and roughly perpendicular to the axis of the camera lens, we assume a linear relationship between the stage and the water surface position in pixel coordinates: Eq. 1, where $H_m$ is the stage in meters, $d_{pixel}$ the water surface position on the picture in pixel.

$$5 \quad H_m = a.d_{pixel} + b \tag{1}$$





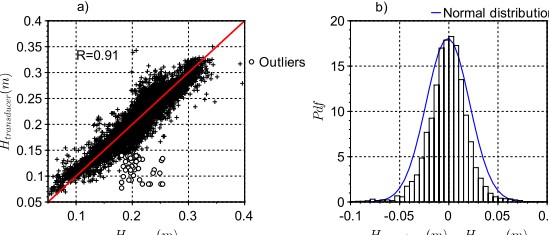
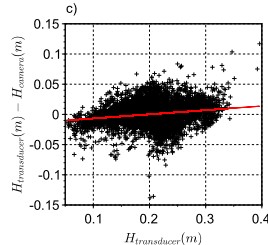

**Figure 9.** Comparison of the transducer data-set ($H_{transducer}$) and the camera data-set ($H_{camera}$) (a) the scatter plot, the cluster of outliers points represented with open symbols has been manually checked and corresponds to wave at the pressure transducer, (b) the error distribution, the normal distribution is set with a mean value of $0.00m$ and a standard deviation of $0.02m$ and (c) the residual plot $H_{transducer}$-$H_{camera}$ regarding $H_{transducer}$, the red line represents the linear regression showing a tilt on the water depth estimation: at high discharges the camera data set underestimates the transducer data set and at low discharges the camera data set overestimates the transducer data set

The slope $a$ of Eq. 1 is given by the millimeters/pixel (mm/px) relationship extracted from the pictures. The camera is fixed, therefore the value is consistent through the entire picture set. Using the board stage on the boulder surface, we get $a = 6mm/px$.

To determine the intercept $b$ of Eq. 1, which is the ground reference of the flow stage, we used part of the stage logger

data-set taking randomly 100 values to extract the intercept $b = -1.6.10^{-3}mm$. This gives a local datum. This water depth calibration is relative the our site and depends on the site particular setting.

## 3.5 Stage validation

Figure 9 (a) shows the comparison between the transducer data-set ($H_{transducer}$) and the camera data-set ($H_{camera}$). The stage prediction from picture analysis is a good estimation of the transducer water level ($R = 0.91$). The water measurement

using the stage board located on the side of the boulder has a lower correlation coefficient($R = 0.71$). The mean value of the difference of the transducer data-set and the camera data-set is $\mu = 0.00m$ and a standard deviation is $\sigma = 0.02m$. Considering a normal distribution (Fig. 9 (b)), the 95 % confidence interval on the error estimation is $[-0.04; 0.04]$ m (the error estimation for the stage board measurement is $[-0.06; 0.06$ m]). Pictures corresponding to the cluster of outliers have been manually checked. Those points correspond to pictures where the water surface is correctly detected but corresponds to waves at the

gauging station or at the pressure transducer.

At very low discharge, boulder clusters emerge near the left bank creating pools and small channels. This channel configuration creates a pond at the water level logger at very low discharge and probably disconnects the water stage measured using the image analysis from that of the pressure transducer. Based on this result the stage measured from image analysis gives a good estimation of the water stage. The high and low frequency variations (i.e. daily or monthly variations) on the transducer

signal are well reproduced by the camera data-set (Fig. 10). The daily snow/ice melt hydrograph which is characteristic of the site, with consistent times of low and high flow each day in the absence of rainstorms, are also shown in Figure 10. While the





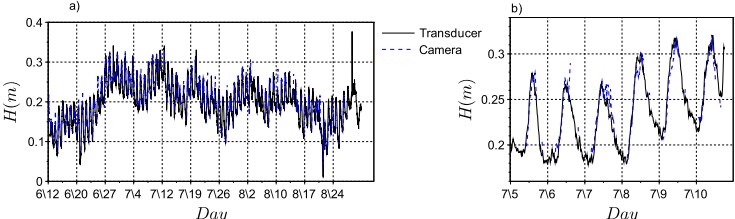

**Figure 10.** Summer 2015 stage time series. The black line represents the transducer data-set ($H_{transducer}$) and the dashed line is the camera data-set ($H_{camera}$). The estimated error on the camera stage relative to pressure transducer is around 3 cm. (a) is from mid June to end of July 2015. The camera data-set fits the transducer data-set. The daily trend as well as the general monthly trend are reproduced (b) from August $5^{th}$ to August $10^{th}$ 2015. At hourly resolution, the trend of the camera stage follows the transducer data closely. However, on rising stage, the camera data-set underestimates the transducer water stage, and on falling stage, the picture data-set slightly overestimates the transducer stage.

results show that the image-based time-lapse method works well, there are some errors that could be reduced. The hypothesis for our stage measurement and the water depth calibration equation is the constant millimeters/pixel (mm/px) relationship over the boulder surface. The underlying assumptions are that the vertical surface is flat and the camera distortion doesn't induce a large variation. Realistically, as the boulder is a natural rock, the vertical face is not exactly vertical and we are not able to

estimate the distortion variation. The probable inconstant mm/px relationship may induce the tilt on the scatter plot (Fig. 9 (a) and (c)). The comparison with the transducer data set shows that at high discharges the camera data set underestimates the transducer data set and at low discharges the camera data set overestimates the transducer data set. The mm/px variations could induce the slightly curved shape of the scatter plot (Fig. 9) but only a better camera resolution (or image scale) would improve the mm/px relationship.

**3.6    Water width measurement**

Using the time-lapse images we also estimated the water surface width. As we did for the stage, width was measured by detecting the threshold between the river and the rocks on both banks. During the field work, flow width was also calibrated on one cross section. For picture analysis, considering the rocks masking the view of the water surface and standing waves in the flow, the measurement cross section was moved about 2 m downstream keeping the same angle across the channel as the

calibrated profile (Fig. 11).

On both sides of the stream some boulders appear at low discharges that are too large to be mobilized by daily high flows. On both rocky areas, a test was done to detect if the rocks had emerged (Test 3 and 4, Fig. 6). If they had, the interrogation area was changed accordingly. On the profile, and as with the water stage, the highest gradient of the grey-scale plot profile was used to detect water edges.





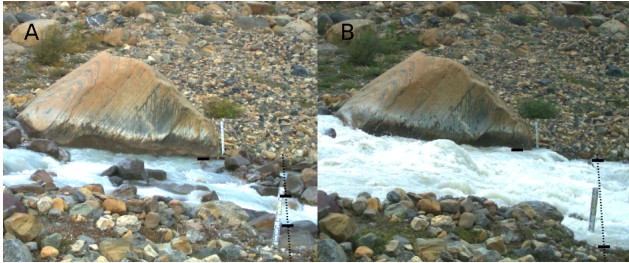

**Figure 11.** The width measurement at low (a) and high (b) discharges. The left line is the water level at the boulder station. The dashed line is the cross section and the right ticks are the detected flow edges.

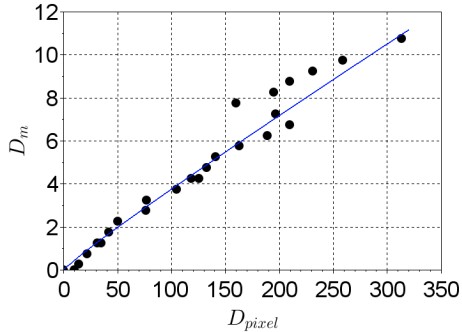

**Figure 12.** The pixel - meter conversion. $D_{pixel}$ is the flow width in pixel and $D_m$ is the flow width in meters. The rating curve was established using a measuring tape across the flow width. The data set has been fit using Eq. 2.

### 3.7 Water width calibration

A measuring tape was extended across the entire Dome stream. The distance across the channel was measured in 0.5 m intervals and the image distance in pixels was converted to meters (Fig. 12). The conversion from distance in pixels to distance in meters is done using Eq. 2, where $D_{pixel}$ is the distance in pixel extracted from the picture analysis and $D_m$ the distance in meters.

5   The measuring tape was not perfectly straight due to the inherent limitations of field work such as the of flow conditions and channel structure, therefore the conversion into meters may be slightly inconsistent, which could induce the shift on Figure 10 around $D_{pixel} = 200$.

$$D_m = 0.05 * (D_{pixel} - 10)^{0.95} + 0.45 \qquad (2)$$

The width measurement faces two principal issues: the water edge detection and the calibration. The width measurement is

10   tightly linked to the boulders on the side of the stream. As the stream widens, the boulders at the channel edge are submerged





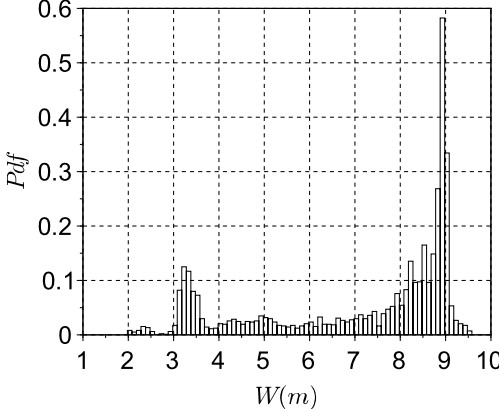

**Figure 13.** The distribution of measured width, The median class ranging from 4 m to 7 m is underrepresented because boulders clusters on each side of the stream make the transition between narrow flows and wide flows fast and nonlinear.

and the grey-scale shift at the edge of the water is less sharp. Inaccurate detection cannot be corrected because of a lack of field validation data for the water width.

The calibration is very sensitive to the camera position. Additional information on camera angle and geometry would increase the calibration accuracy and improve the width measurement. Furthermore, moving the cross section a small distance away

from the calibrated cross section induced some error on the conversion length in meters from length in pixel. Nevertheless with some refinement, as the distortion of the picture (due to the angle, camera setting which defines the pixel/meters conversion) is a monotone function there is no major effect on the relative width variations.

### 3.8   The width observations

At very low discharges the stream bed is covered by large boulders clearly seen in the pictures, rocks are fully submerged at high

flows, creating large surface waves. The transition between low and high flows creates secondary channels and we chose to only consider the main channel and not the side channels at low and medium flows. This induces an underestimation of the width at low discharges. The flow widening is also strongly impacted by those boulder clusters, at low discharge the main channel is contained in the center of the bed, and water stage has to be quite high to be over both clusters. The transition between wide and narrow channels is fast and therefore intermediate widths (between 4 and 7 meters wide) are underrepresented in the width

data (Fig. 13). The image information reveals these aspects of the hydraulics of the channel (and that affect stage changes) that would not be known with stage data alone.





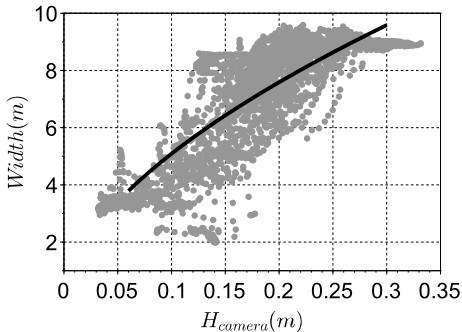

**Figure 14.** The water surface width as a function of the water height. The stage is taken from the camera data-set. The flow width is taken from the width detection and Eq. 2. The trend line equation is $W = 19.19 H_{camera}^{0.58}$.

### 3.9 The width stage relationship

In braided channel studies the wetted surface measurement has been used as a substitute for stage to estimate the discharge (Smith et al., 1996; Ashmore and Sauks, 2006; Gleason et al., 2015). Previous studies have shown the correlation between the wetted surface and the discharge with an exponent from 0.5 (Smith et al., 1996) to 1 (Ashmore and Sauks, 2006). The

width response to discharge change are much higher in these braided channels than in many other streams and this gives the potential for using width in addition to, or instead of, stage changes as the primary variable for estimating discharge. In the Dome stream case, although it is not a braided stream, the relatively shallow cross-section also gives significant widening of flow with increasing stage and the positive trend is clear with an exponent close to 0.6 (Fig. 14). However, the trend is not linear and data scatter is quite large because of the irregular geometry of the cross-section, and the bouldery channel edges (especially

at low discharge). Nevertheless, the width estimation could be a reliable approximation to the stage measurement. Even if not used directly as a discharge surrogate, the width data give additional information on the hydraulic geometry of the channel that would be difficult to predict theoretically for this type of channel. The width data also reveal some interesting hysteresis in the flow hydraulics. On 67% of the 81 daily flow peaks on the Dome river for which there are data, the mean value of the stream width over the water stage range $]0.5.H_{max}, H_{max}]$, $H_{max}$ being the daily maximum water stage, is higher on the falling limb

of the daily hydrograph than on the rising limb. The width increase ranges from 0.5% to 26 % with a mean value of 13.9% and a median value of 9.1%.

     This produces an obvious hysteresis loop in the width-stage plot (Fig. 15), as can often be found in the stage-discharge relationship (Petersen-Øverleir, 2006). This adds further information for interpretation of the flow and reveals an important aspect of the channel hydraulics that may be the result of the complexity of the flow over and around the boulder clusters, so

that the macro-roughness has an important effect on the flow detected from the image analysis.





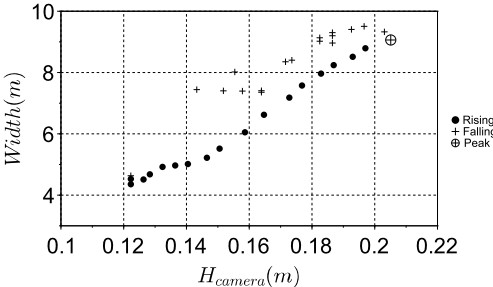

**Figure 15.** The water surface width as a function of water stage, for falling flows (+), rising flows (•) and the discharge peak (⊕). The curve has a loop, for the same water stage the flow is wider on a falling flow than on a rising flow. The end of the falling flows usually happen at night, therefore the data are missing.

## 4    Discussion

The method described in this paper is similar to two recent studies proposed by Gleason et al. (2015); Young et al. (2015), but it differs in both general approach, field data and image selection, and processing. Young et al. (2015) assumed a V-shape of their studied cross section, so that the edge coordinate is linearly related to the water stage. They estimate the water level

from the water edge, without on-site validation data and use a statistical estimation procedure combined with assumed channel geometry to derive water level changes. In the Dome stream case and with width measurement, the water stage and water width are not linearly related, which gives information on the stream cross section despite the lack of topographic survey and shows that assumptions of the type used by Young et al. (2015) would not be reliable in this case. It also points to the difficulty of reliably predicting flows using a standard resistance assumption in this type of channel. In addition Young et al. (2015) use

manual methods to identify water edges on all images. Gleason et al. (2015) focus on water area detection in a large braided channel and not on direct water stage measurements or on small, steep channels.

The environmental conditions (e.g. sun position, fog, rain) are the main common difficulties that reduce the picture quality and make picture filtering an important step in the process. Gleason et al. (2015); Young et al. (2015) identify similar issues but adopt different approaches. Young et al. (2015) use manual image selection in contrast to our automated selection procedures

which makes it possible to process a much larger image set and derive much higher frequency data (15 minutes vs. 4 hours). Gleason et al. (2015) adopt semi-automated image procedures which differ in detail from ours but their procedures result in either retention or rejection of images for measurement whereas we derive alternative detection criteria (cross section resizing, peak detection... see Table 1 ) for a subset of images rather than eliminating them completely from the data set. Consequently are able to retain much higher frequency monitoring relative to Gleason et al. (2015). Field data on the site characteristics




avoids having to make assumptions about the site such as the cross section shape (Young et al., 2015) or working without any ground data for validation (Gleason et al., 2015).

Improved image acquisition is the key component for improving remote sensing accuracy and time coverage. The use of inexpensive time-lapse cameras introduces some limitations that can be mitigated. A higher image resolution and a better

camera position (reducing sunlight effects, or improving the position relative to the boulder face for example) would improve the measurement accuracy for both the water stage and the channel width. These refinements are easy to implement and test.

Another obvious limitation is the restriction to day time images. In the Dome case, night and twilight represent roughly 1/3 of the day in the summer meltwater period for which data are needed. Using a night vision camera may extend the effective monitoring times but we have not tested this. The limitation may be less significant if only certain flow information is needed

rather than a 24 hour continuous signal. Even without continuous data useful information on channel hydraulics can also be obtained from this type of monitoring. These procedures and image processing steps may be changed to fit site characteristics or data needs. In this case the method provided the necessary seasonal stage signal and timing of daily peaks needed for the study objective of comparison between the ungauged tributary and the main channel flow.

## 5  Conclusions

The results demonstrate the effectiveness of a simple measurement apparatus for flow stage and water surface width: low-cost time-lapse camera and a few simple field measurements. Fully automatic image processing to select images and to detect the water level and edges makes it possible to process a large number of images to produce a long, high temporal resolution, data set. It shows that reliable water stage and water width measurement can be measured at a small (minutes) time steps over 3 months in this case. The estimated hydraulic parameters reliably reproduce the hourly, daily and monthly variation in flow of

this pro-glacial river compared to pressure-transducer stage data. The low cost of the camera (approximately $600) and the very easy data collection makes the image processing a powerful tool for this type of river monitoring especially on small headwater streams. Image analysis produced a larger variety of data and information than a simple water stage transducer alone can yield. Indeed pictures provide visible data such as weather conditions (snow cover, freezing conditions, rain), and water surface conditions (surface waves, eddies, jumps) and details of the flow hydraulics and geometry over the full range of

discharge. Image analysis can also be extended to other hydraulic measurements such as the water slope. The method extends the available methods for inexpensive terrestrial remote sensing of river flow at high frequency and extended time periods applicable especially to small channels with complex flow.

*Acknowledgements.* This research was supported by a Natural Sciences and Engineering Research Council of Canada Discover Grant to Ashmore. We thank Sarah Peirce, Lara Middleton and Matilde Welber for field work assistance and Tobi Gardner for help with the water

level logger.



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
