# Peer review of "Technical note: Stage and water width measurement of a mountain stream using a simple time-lapse camera"

_Hydrology and Earth System Sciences, 2017_

## Referee Comment (RC1) · Anonymous Referee #1 · 8 Aug 2017

General comment:

This technical note provides the imaging approach to quantify the width and level of running water in a mountainous area using a commercially available time-lapse camera. To categorize the image quality/characteristics, the image taken by the camera was analyzed using the (eight) target zones, and this point differed from the approaches described in Gleason et al. (2015) and Young et al. (2015). Utilization of image to quantify the hydrological parameters is active topics. The manuscript provides the experience of the authors and this is potentially beneficial to strengthen our understanding of hydrological processes. In my view, the manuscript is worthy of publication after

some modification and clarification.

Specific comments:

The caption of Figure 3, the locations of the pressure transducer is better to be specified in the figure.

Page 5. line 4 and others, Authors used 'sorted' but I suggest to use 'classify' or 'categorize' since 'sorted' has another meaning, the arrangement of data in a prescribed sequence.

Page 5, line 18 and following paragraph. It is better to refer 'Test 1' and 'Test 2' in the main text as like did for Test 3 and Test 4 to make clearer the link between the discussion in the main text and the contents of Figure 6.

Figure 7: Please explain the meaning/definition of red and blue lines.

Page 13, line 9, I think it is better to insert some conjunction before 'rocks are fully...'

Page 14, line 14, I can not understand the meaning of ']0.5.Hmax,Hmax],Hmax'

Page 14, lines 18-20. This sentence is not clear. Please rephrase.

Page 15, line 13 'make picture filtering an important step in the process' is not clear. Please rephrase.

Page 16, line 6, the importance of camera angle was considerably discussed in Tsubaki et al. (2011) and please consider to cite here.

Reference suggestions (not must but suggested to refer) (1) Stumpf, A., E. Augereau, C. Delacourt, and J. Bonnier (2016), Photogrammetric discharge monitoring of small tropical mountain rivers: A case study at Rivière des Pluies, Réunion Island, Water Resour. Res., 52, 4550–4570,

(2) Ran, Q., Li, W., Liao, Q., Tang, H., and Wang, M. (2016) Application of an automated LSPIV system in a mountainous stream for continuous flood flow measurements. Hydrol. Process., 30: 3014–3029. doi: 10.1002/hyp.10836.

Technical corrections:

Page 1. line 19, "Ground-based remote sensing 'increasingly' is providing", 'increasingly' seems improper here.

Page 2, line 24, 'oblique time-lapse oblique image' There are two oblique and please rephrase the sentence.

Page 4, line 12, 1536 by 2048 would be 2048 by 1536. '.jgp' would be '.jpg'.

Page 5, lines 8-9. Remove space between 'estimation' and ':'. The parentheses '(' before Figs 4(a) to 4(c) were not closed by corresponding ')' so please edit the sentence.

Figure 5, Change color of box and text to increase the contrast between the background picture and additional information.

Page 10, lines 3, 5, 11 and 12: Insert small space between number and unit. The unit should be the regular font, not the italic typeface. Remove space before '%'.

Figure 10. The dashed blue line and the solid black line are difficult to distinguish. Please arrange the figure to be easier to distinguish two lines. In the caption, 'August' may be 'July'.

Figure 11, Lines in figures are difficult to identify. Please change color or/and style to highlight the information.

---

## Referee Comment (RC2) · D. Young (Referee) · 18 Aug 2017

This paper builds on previous work to further demonstrate the value of an inexpensive, portable and easily-sited time-lapse camera for making frequent (potentially continuous) measurements of stage and water width. The system is applied to a mountain stream with an irregular bed, where other technologies might be difficult to deploy. Particular contributions of this paper include: (a) the validation of an image-based system against a pressure-based system for measuring stage; (b) automation of detection of the water edge in the images; (c) observations of the detailed relationship between stage as measured on the rock surface and as measured by pressure; (d) a demonstration of the successful deployment of this novel type of system in a new site.

The work is closely related to that reported in Young et al. (2015). (I am the first author of that paper.) Our paper lacked an independent measure for comparison, and this paper addresses this problem and provides interesting (and reassuring) results. The image-based approach has thus been validated using a separate measurement, which we were unable to do. The authors take the data further, observing some phenomena such as differences depending on whether the flow is increasing or decreasing, which merit further investigation in the future.

I have two relatively minor criticisms. The first is that the significance of automation of water edge detection is a little overstated; the second is that stage and width measurements are treated as separate problems, rather than integrated into a single model.

The approach depends on detection of the water/rock boundary in the images. It is important to automate this if large amounts of data are to be processed (and one can imagine a study in which many cameras are deployed over an extended period, all collecting many images per hour). However, Leduc et al.'s method depends on having a large rock surface such that the water surface lies across it for all stages of interest, and where the water/rock boundary exhibits the highest contrast (or second-highest in certain lighting conditions). Given such a situation, our method could also have been fully automated - we used manual intervention to select the correct edge in a much more cluttered image, using a selection of smaller rocks. Leduc et al.'s images show strong contrast between the water and the rock (the water is very bright in the images that appear in the paper) and it is not clear that their method will work well in situations where the water is flowing more smoothly, or the river channel is more complex. Their image analysis algorithm has some ad hoc elements which may not generalise well.

It would be worth noting that selection of the maximum grey-level gradient along a vertical profile is almost the same as Canny edge detection in a narrow strip - the only real difference is that Canny uses some Gaussian smoothing to reduce noise. Thus

both this paper and ours use (not surprisingly) rather similar image analysis techniques.

I think it would be useful to include some more discussion of the tradeoff between automatic analysis and the need for careful site selection, and also of the limitations of using a single surface for the stage measurement.

Our approach of combining measurements on multiple surfaces (both near-vertical and near-horizontal) allows for a larger range of stages to be covered and for estimates to be made of the consistency and statistical reliability of the measurements, as well as providing a rough estimate of the channel geometry. Leduc et al.'s approach relies on a more carefully chosen camera position and a suitable large rock, which allows automated image processing and a simpler analysis of the image measurements. In my view, the merits of the two approaches are complementary, and future work should draw on both.

The water width measurement is interesting, but again I am concerned about how well this would generalise to other settings. A little more needs to be said about how the nonlinear fit in Eq 2 was arrived at - is this a purely empirical equation or is there a model behind it? The final paragraph of section 3.7 does, however, provide a good summary of the issues.

Section 3.9 is novel and interesting, going significantly beyond our work, and reveals how the technique can give results that would not be available in any other way.

I feel that section 4 of the paper raises interesting questions, and I agree strongly with the conclusions reached in section 5.

Overall, I think this is a very strong contribution and I am happy to recommend publication.

Technical comments:

p 1, l 17: Extra parentheses in reference (cf l 21)

p 2, l 6: We didn't know the channel geometry - we assumed a V-shaped section and estimated the slopes from the data.

p 2, l 12: boulder -> bouldery

p 2, l 24: too many 'oblique's

p 4, Fig 3, and subsequent figures: can the figures be made bigger? When the paper is printed, the details are hard to see. This is particularly the case for Figs 9, 10 and 11, and the central element of Fig 1.

p 6, l 1-14: This section could be a little and more explicit. I didn't understand how the re-sizing operation worked, or what its basis is.

Table 1: the values for SDs assume, I think, a brightness range of 0-255. This ought to be stated explicitly (it's common, but not universal).

p 10, l 2: Should "board stage" be "stage board" for the calibration? If not, I don't understand what is meant.

p 14, l 5: are much higher -> is much higher

p 14, l 14: First bracket backward

―――――――――――――――――

---

## Referee Comment (RC3) · Anonymous Referee #3 · 24 Aug 2017

This paper seeks to define a methodology for retrieving pertinent river variables from time lapse imagery. At its core, this methodology relies on site selection to be effective: the authors were quite clever to notice different image regions and exploit these for filtering. This is a great idea that could be adopted to other studies. The authors also rely heavily on a 'nearly vertical' boulder for stage and width measurements. This is also rather clever, but acts as a double edged sword: What if no boulder can be found with a flat face orthogonal to the camera plane? The authors need to elaborate, at paragraph length or longer, the considerations of site selection for this method. Is it serendipity that the image regions and boulders emerged, or were they chosen? In addition, the authors MUST quantify their method much more accurately- descriptions

of quantities are too vague and not replicable. If I want to apply this method and need to find classification regions, what distributions of intensity should I seek? How should I define my different image regions, or will this method only work at this one site with this unique assemblage of morphologies? Therein lies my main concern- the clever filtering techniques that are an improvement on Gleason et al may not work everywhere. The authors need to more clearly define their threshold values, and I think distributions of intensity in each of the eight regions should be shown at multiple flow levels to prove their utility beyond taking the authors' word for it. Precise values are given in Table 1, but not in the text, and no distributions are shown. The reported wave issues are also troubling for the site selection issues discussed above. How much of an issue would these be in a different site? The photogrammetry here is effective, and I do not quibble with the empirical functions for stage and width. However, these do require calibration that makes this procedure time consuming. Could the authors comment on the amount of time needed to make measurements for sufficient calibration? i.e. could I set up 50 of these stations, all well-calibrated, in a summer in remote terrain? This is an important discussion left out of the manuscript that should be added, as the authors propose that this method should be adopted for such streams. The use of precise rotation matrix measurements and classic photogrammetry would obviate the need for this calibration, as would establishing several cameras in stereo. Overall, I recommend this paper for publication, provided the authors write several new sections detailing the concerns discussed above- all of which relate to the role of site selection in this method. Some minor comments: The English writing is sloppy at times- needless plurals, .jgp instead of .jpg, backward brackets, etc. This needs to be amended and made more professional. Page 4, line 13: What is this standard deviation? It is insufficient to say 'very low.' Also, this is a filter, correct? It should be identified as such. Page 4 -> page 5: This writing is redundant- section 3.1 should be eliminated and combined into section 3.2 Page 5, line 15+: again, what is the threshold of SD? Also, this writing is unclear-words like 'high' and 'drops' and 'smoother' are used, which are imprecise. Since this paper proposes an algorithm for installation and monitoring, it must be specific so as

to be reproduced.

---

## Author Comment (AC1) · 10 Oct 2017

10 octobre 2017
* * *
We thank the three referees for their comments. Our responses below are organised to respond to each review in sequence. Where the response is that we agree with suggested minor edits we have indicated that we will make those the changes in the final version of the paper. The focus of this response is on the more substantial comments.

*Answers are in italic font.*
* * *
**Answers to : Anonymous Referee 1**

**General comment :**

This technical note provides the imaging approach to quantify the width and level of running water in a mountainous area using a commercially available time-lapse camera. To categorize the image quality/characteristics, the image taken by the camera was analyzed using the (eight) target zones, and this point differed from the approaches described in Gleason et al. (2015) and Young et al. (2015). Utilization of image to quantify the hydrological parameters is active topics. The manuscript provides the experience of the authors and this is potentially beneficial to strengthen our understanding of hydrological processes. In my view, the manuscript is worthy of publication after some modification and clarification.

**Specific comments :**

The caption of Figure 3, the locations of the pressure transducer is better to be specified in the figure.

*Agreed*

Page 5. line 4 and others, Authors used 'sorted' but I suggest to use 'classify' or 'categorize' since 'sorted' has another meaning, the arrangement of data in a prescribed sequence.
*Agreed*

Page 5, line 18 and following paragraph. It is better to refer 'Test 1' and 'Test 2' in the main text as like did for Test 3 and Test 4 to make clearer the link between the discussion in the main text and the contents of Figure 6.
*Agreed*

Figure 7 : Please explain the meaning/definition of red and blue lines.
*The 2 lines represent 2 different water stages. Caption will be clarified.*

Page 13, line 9, I think it is better to insert some conjunction before 'rocks are fully...'
*Agreed*

Page 14, line 14, I can not understand the meaning of ']0.5.Hmax,Hmax],Hmax'
*The sentence will be rephrased to clarify this.*

Page 14, lines 18-20. This sentence is not clear. Please rephrase.
*Will be done*

Page 15, line 13 'make picture filtering an important step in the process' is not clear. Please rephrase.
*Will be done*

Page 16, line6, the importance of camera angle was considerably discussed in Tsubaki et al. (2011) and please consider to cite here.
*We would appreciate having the full reference here so that we can locate this paper.*

Reference suggestions (not must but suggested to refer) (1) Stumpf, A., E. Augereau, C. Delacourt, and J. Bonnier (2016), Photogrammetric discharge monitoring of small tropical mountain rivers : A case study at Rivière des Pluies, Réunion Island, Water Resour. Res., 52, 4550–4570 (2)Ran,Q.,Li,W.,Liao,Q.,Tang,H.,and Wang,M.(2016)Application of anautomated LSPIV system in a mountainous stream for continuous flood flow measurements.

*Thank you for the citation suggestions. These can be added.*

Technical corrections :

*These will all be done*

Page 1. line 19, "Ground-based remote sensing 'increasingly' is providing", 'increasingly' seems improper here.

Page 2, line 24, 'oblique time-lapse oblique image' There are two oblique and please rephrase the sentence.

Page 4, line 12, 1536 by 2048 would be 2048 by 1536. '.jgp' would be '.jpg'.

Page 5, lines 8-9. Remove space between 'estimation' and ' :'. The parentheses '(' before Figs 4(a) to 4(c) were not closed by corresponding ')' so please edit the sentence.

Figure 5, Change color of box and text to increase the contrast between the background picture and additional information.

Page 10, lines 3, 5, 11 and 12 : Insert small space between number and unit. The unit should be the regular font, not the italic typeface. Remove space before '%'.

Figure 10. The dashed blue line and the solid black line are difficult to distinguish. Please arrange the figure to be easier to distinguish two lines. In the caption, 'August' may be 'July'.

Figure 11, Lines in figures are difficult to identify. Please change color or/and style to highlight the information.

**Answers to : D. Young (Referee)**

This paper builds on previous work to further demonstrate the value of an inexpensive, portable and easily-sited time-lapse camera for making frequent (potentially continuous) measurements of stage and water width. The system is applied to a mountain stream with an irregular bed, where other technologies might be difficult to deploy. Particular contributions of this paper include : (a) the validation of an image-based system against a pressure-based system for measuring stage ; (b) automation of detection of the water edge in the images ; (c) observations of the detailed relationship between stage as measured on the rock surface and as measured by pressure ; (d) a demonstration of the successful deployment of this novel type of system in a new site. The work is closely related to that reported in Young et al. (2015). (I am the first author of that paper.) Our paper lacked an independent measure for comparison, and this paper addresses this problem and provides interesting (and reassuring) results. The image-based approach has thus been validated using a separate measurement, which we were unable to do. The authors take the data further, observing some phenomena such as differences depending on whether the flow is increasing or decreasing, which merit further investigation in the future. I have two relatively minor criticisms. The first is that the significance of automation of water edge detection is a little overstated ; the second is that stage and width measurements are treated as separate problems, rather than integrated into a single model. The approach depends on detection of the water/rock boundary in the images. It is important to automate this if large amounts of data are to be processed (and one can imagine a study in which many cameras are deployed over an extended period, all collecting many images per hour). However, Leduc et al.'s method depends on having a large rock surface such that the water surface lies across it for all stages of interest, and where the water/rock boundary exhibits the highest contrast (or second-highest in certain lighting conditions). Given such a situation, our method could also have been fully automated - we used manual intervention to select the correct edge in a much more cluttered image, using a selection of smaller rocks. Leduc et al.'s images show strong contrast between the water and the rock (the water is very bright in the images that appear in the paper) and it is not clear that their method will work well in situations where the water is flowing more smoothly, or the river channel is more complex. Their image analysis algorithm has some ad hoc elements which may not generalise well. It would be worth noting that selection of the maximum grey-level gradient along a vertical profile is almost the same as Canny edge detection in a narrow strip - the only real difference is that Canny uses some Gaussian smoothing to reduce noise. Thus both this paper and ours use(not surprisingly)rather similar image analysis techniques. I think it would be useful to include some more discussion of the trade off between automatic analysis and the need for careful site selection, and also of the limitations of using a single surface for the stage measurement. Our approach of combining measurements on multiple surfaces (both near-vertical and near-horizontal) allows for a larger range of stages to be covered and for estimates to be made of the consistency and statistical reliability of the measurements, as well as providing a rough estimate of the channel geometry. Leduc et al.'s approach relies on a more carefully chosen camera position and

a suitable large rock, which allows automated image processing and a simpler analysis of the image measurements. In my view, the merits of the two approaches are complementary, and future work should draw on both. The water width measurement is interesting, but again I am concerned about how well this would generalise to other settings. A little more needs to be said about how the nonlinear fit in Eq 2 was arrived at - is this a purely empirical equation or is there a model behind it ? The final paragraph of section 3.7 does, however, provide a good summary of the issues. Section 3.9 is novel and interesting, going significantly beyond our work, and reveals how the technique can give results that would not be available in any other way. I feel that section 4 of the paper raises interesting questions, and I agree strongly with the conclusions reached in section 5. Overall, I think this is a very strong contribution and I am happy to recommend publication.

*Thank you for your comments. As you can tell, your paper was a help to us although we discovered it only after we had done our field work. The general comment on needing a vertical rock surface is a slight misunderstanding that we can clarify in the revisions. Our original intent was to use the stage boards which could be installed at almost any site. The stage boards worked but in the analysis we discovered that the boulder face, a serendipitous feature of the site which we calibrated while we were there, actually gave us slightly better results. The boulder face is not a requirement for the method we applied – any vertical calibrated surface could be used. Further testing by others in different environments would help the community to assess how widely applicable this approach is. Thank you also for giving more detail on how the two methods relate to each other and pointing out the matter of whether a single vertical or a length of channel is preferable. We can revise the discussion section to draw this out more. We agree also that more work on width (and other hydraulic parameters) is needed and the edge detection method from your paper may be better suited to this for some channels.*

*Equation 2 is an empirical fit.*

Technical comments :

*Thank you for these. We will make all of these changes in the final paper.*

p 1, l 17 : Extra parentheses in reference (cf l 21)

p 2, l 6 : We didn't know the channel geometry - we assumed a V-shaped section and estimated the slopes from the data.

p 2, l 12 : boulder -> bouldery

p 2, l 24 : too many 'oblique's

p 4, Fig 3, and subsequent figures : can the figures be made bigger ? When the paper is printed, the details are hard to see. This is particularly the case for Figs 9, 10 and 11, and the central element of Fig 1.

p 6, l 1-14 : This section could be a little and more explicit. I didn't understand how the re-sizing operation worked, or what its basis is.

Table 1 : the values for SDs assume, I think, a brightness range of 0-255. This ought

to be stated explicitly (it's common, but not universal)

    p 10, l 2 : Should "board stage" be "stage board" for the calibration ? If not, I don't understand what is meant.

    p 14, l 5 : are much higher -> is much higher

    p 14, l 14 : First bracket backward

**Answers to : Anonymous Referee 3**

This paper seeks to define a methodology for retrieving pertinent river variables from time lapse imagery. At its core, this methodology relies on site selection to be effective : the authors were quite clever to notice different image regions and exploit these for filtering. This is a great idea that could be adopted to other studies. The authors also rely heavily on a 'nearly vertical' boulder for stage and width measurements. This is also rather clever, but acts as a double edged sword : What if no boulder can be found with a flat face orthogonal to the camera plane ?

*As in our response to the second reviewer : the rock face was fortunate but the method works for stage boards (or other surfaces) installed at a site (which was the original intent), so the method is not reliant on the boulder surface. Any calibrated near-vertical surface will work. The flat boulder is indeed convenient, but in the paper, we also show satisfactory the result using one of the stage boards. The results could be refined and made more precise for particular installation.*

The authors need to elaborate, at paragraph length or longer, the considerations of site selection for this method. Is it serendipity that the image regions and boulders emerged, or were they chosen ? In addition, the authors MUST quantify their method much more accurately- descriptions of quantities are too vague and not replicable. If I want to apply this method and need to find classification regions, what distributions of intensity should I seek ? How should I define my different image regions, or will this method only work at this one site with this unique assemblage of morphologies ?

*We illustrate the method for the particular site for which we sought a stage record. We did not choose the site for its particular characteristics, we were trying to get a stage record for the site given the characteristics that it had. We are not sure what is vague in our descriptions. One need not seek particular site characteristics for this method, but the image selection criteria would need to be customized for the site. This is partly a 'trial and error' method but once established for a site, which is not a lengthy process, it is automated. The threshold issue is the key point of our picture classification. However, we don't want to focus too much on the 4 threshold settings for this particular site as they would need calibration for a different site. Our focus was to show the method. We can add an appendix with the threshold curves if this would be useful (see below, figure 1).*

Therein lies my main concern-the clever filtering techniques that are an improvement on Gleason et al may not work everywhere. The authors need to more clearly define their threshold values, and I think distributions of intensity in each of the eight regions should be shown at multiple flow levels to prove their utility beyond taking the authors' word for it. Precise values are given in Table 1, but not in the text, and no distributions are shown. The reported wave issues are also troubling for the site selection issues discussed above. How much of an issue would these be in a different site ?

[Figure]

(a) Boulder shadow - Test 2

(b) Exposed rocks - Test 3

[Figure]

(c) Light snow cover - Test 4

FIGURE 1 – The different distribution used to set the thresholds

*We are not sure what is being requested here. The filtering will need to be customized for a particular site (as it was in Gleason et al.) but the process is general. As the comment says, we have given values in Table 1, so we are not sure why repetition in the text is necessary. Note that the image regions are well outside the channel so are not affected by flow level except in the case of removing mid-channel rocks from the width estimation which would not be an issue in channels with large relative depth. We are not sure what the concern is with waves. We have explained this feature of the flow and one would need to be aware of it, but the stage signal can be filtered to account for these if necessary (as it is in any stage recording in an open channel) and we show coherence between the open water level and the logger level in a pipe.*

The photogrammetry here is effective, and I do not quibble with the empirical functions for stage and width. However, these do require calibration that makes this procedure time consuming. Could the authors comment on the amount of time needed to make measurements for sufficient calibration? i.e. could I set up 50 of these stations, all well-calibrated, in a summer in remote terrain? This is an important discussion left out of the manuscript that should be added, as the authors propose that this method should be adopted for such streams. The use of precise rotation matrix measurements and classic photogrammetry would obviate the need for this calibration, as would establishing several cameras in stereo. Overall, I recommend this paper for publication, provided the authors write several new sections detailing the concerns discussed above-all of which relate to the role of site selection in this method.

*We are not suggesting what should be adopted, only showing another possibility for flow monitoring. Time needed for installation would obviously depend on local circumstances. The physical installation and calibration can be done at a site in 1-2 hours without a pressure transducer using two people (possibly less with experience). We did not think of whether this was useful for a large monitoring network but it is clearly a feasible option for selected research sites. Post-processing to apply image selection criteria can be done in a few hours and with experience some criteria (e.g. dark, sun, water droplets etc.) may be generalizable. We agree that stereo applications are another option, as are PTV applications, but we sought a simple low-cost option equivalent to standard stage recording. We would be pleased to see somebody develop the photogrammetric approaches proposed. We can add these points to the discussion.*

Some minor comments :

*Thank you. We will make the minor corrections*

The English writing is sloppy at times- needless plurals, .jgp instead of .jpg, backward brackets, etc. This needs to be amended and made more professional.
*The backward brackets are deliberate to indicate the semi open interval, we didn't include 0.5.Hmax in our interval. We will, of course, proofread the final text for minor*

*errors.*

Page 4, line 13 : What is this standard deviation ? It is insufficient to say 'very low.' Also,this is a filter,correct ? It should be identified as such.

*The standard deviation is here referring to the standard deviation of the picture (or of a cropped area of the picture) of the grey scale which indicates the noise of the picture. A standard deviation of 0 indicates a one color picture while a high standard deviation (eg > 50 on our site) indicates a very large range of color of the picture. The 'very low' threshold is easy to set, the darkness makes the color pretty much even on the picture surface in contrast to daylight when the different color shades make the standard deviation high. The definition of 'low' and 'high' is site dependent. Indeed, the standard deviation of a "normal" weather condition depends on the river background, the picture setting (if there is a lot of sky on the picture or not), rocks color... The 'very low' threshold has to be set relative to the site and the picture components.*

Page4->page5 : This writing is redundant- section 3.1 should be eliminated and combined into section 3.2.

*In those two sections we want first to explain all the issues we have to face and secondly we want to present the different tests we made to get around those issues. This clarification was requested by the associate editor prior to distribution to referees.*

Page 5, line 15+ : again, what is the threshold of SD ? Also, this writing is unclear words like 'high' and 'drops' and 'smoother' are used, which are imprecise. Since this paper proposes an algorithm for installation and monitoring, it must be specific so as to be reproduced.

*We are not sure why these are problematic. We are only descriptively establishing a sense of the direction of change or difference. See our response above to the issues of image selection criteria and site selection and image standard deviation.*